# Activation of the same mGluR5 receptors in the amygdala causes divergent effects on specific versus indiscriminate fear

Mohammed Mostafizur Rahman[1,2†], Sonal Kedia[1†], Giselle Fernandes[1†], Sumantra Chattarji[1,2,3*]

[1]National Centre for Biological Sciences, Tata Institute of Fundamental Research, Bangalore, India; [2]Centre for Brain Development and Repair, Institute for Stem Cell Biology and Regenerative Medicine, Bangalore, India; [3]Centre for Integrative Physiology, Deanery of Biomedical Sciences, University of Edinburgh, Edinburgh, United Kingdom

**Abstract** Although mGluR5-antagonists prevent fear and anxiety, little is known about how the same receptor in the amygdala gives rise to both. Combining in vitro and in vivo activation of mGluR5 in rats, we identify specific changes in intrinsic excitability and synaptic plasticity in basolateral amygdala neurons that give rise to temporally distinct and mutually exclusive effects on fear-related behaviors. The immediate impact of mGluR5 activation is to produce anxiety manifested as indiscriminate fear of both tone and context. Surprisingly, this state does not interfere with the proper encoding of tone-shock associations that eventually lead to enhanced cue-specific fear. These results provide a new framework for dissecting the functional impact of amygdalar mGluR-plasticity on fear versus anxiety in health and disease.

*For correspondence: shona@ncbs.res.in

†These authors contributed equally to this work

Competing interests: The authors declare that no competing interests exist.

## Introduction

The group I metabotropic glutamate receptor subtype mGluR5 in the amygdala plays an important role in conditioned fear and anxiety-like behavior. For instance, a detailed study spanning multiple levels of analysis demonstrated that administration of a specific mGluR5 antagonist into the lateral nucleus of the amygdala (LA) impairs the acquisition of auditory fear conditioning (*Rodrigues et al., 2002*). Further, using in vitro electrophysiological experiments, this mGluR5 antagonist was shown to impair long-term potentiation (LTP) at thalamic inputs to the LA (*Rodrigues et al., 2002*). Interestingly, intra-amygdaloid microinjections of the same antagonist also prevents anxiety in a variety of rodent models (*La Mora et al., 2006*). Consistent with these animal studies, mGluR5 antagonists have also been reported to act as effective anxiolytics in human conditions of fear and anxiety (*Porter et al., 2005*). Although these studies collectively implicate mGluR5 activity in the amygdala, they do not explain how activation of the same receptor gives rise to both fear and anxiety (*La Mora et al., 2006*; *Rodrigues et al., 2002*). A complementary experimental strategy, involving selective *activation* of the same receptor, may offer a way of dissecting the neuronal basis of these two amygdala-dependent behaviors. But this line of investigation remains largely unexplored in the amygdala, as much of our current understanding is based primarily on studies that used systemic administration of mGluR5 antagonists to modulate these behaviors in rodents (*Swanson et al., 2005*).

In contrast to the amygdala, a growing body of evidence from the hippocampus on mGluR5-dependent synaptic plasticity has emerged from electrophysiological experiments using in vitro application of a specific agonist (*RS*)−3,5-dihydroxyphenylglycine (DHPG) (*Malenka and Bear, 2004*). For instance, in area CA1 of the hippocampus, in vitro treatment with DHPG causes long-

term depression (LTD) of synaptic strength (*Huber et al., 2000*), as well as reduction in the frequency of miniature excitatory postsynaptic currents (*Snyder et al., 2001*). This raises interesting questions about the effects of similar manipulations in the amygdala. An earlier study (*Rudy and Matus-Amat, 2009*) showed that infusion of DHPG into the basolateral amygdala (BLA) increased freezing induced by fear conditioning using a weak foot shock. However, this study did not examine the potential impact of such in vivo manipulations on anxiety or fear generalization. Further, the underlying cellular and synaptic mechanisms were not explored in this study. Since mGluR5 antagonists block LTP in the LA (*Rodrigues et al., 2002*), would activation of mGluR5 using DHPG induce LTP in the LA? Would it also have the opposite effect on miniature excitatory postsynaptic currents compared to the hippocampus? If the cellular effects of mGluR5 activation are indeed different in the amygdala, what are its behavioral consequences? Since mGluR5-dependent LTP in the LA has been linked to fear conditioning, would selective activation of this receptor only facilitate the formation of fear memories? Or would it also affect anxiety-like behavior? Importantly, would such pharmacological manipulations provide a way to distinguish between the two behavioral states?

## Results

To address these questions, we first examined the impact of bath application of DHPG on principal neurons of the LA using whole-cell current-clamp recordings in coronal brain slices prepared from adult male rats (*Figure 1a*). As reported earlier (*Faber et al., 2001*), LA principal neurons show spike-frequency adaptation upon somatic injection of depolarizing currents (*Figure 1b*). For increasing values of current injected, the same cell fired more action potentials in the presence of DHPG relative to baseline levels (*Figure 1c*), indicative of enhanced postsynaptic excitability. DHPG has also been shown to reduce AMPAR-mediated synaptic transmission in the hippocampus (*Fitzjohn et al., 2001*; *Snyder et al., 2001*). Hence, we next examined the effects of DHPG on AMPAR-mediated spontaneous excitatory postsynaptic currents (sEPSCs), as well as miniature excitatory postsynaptic currents (mEPSCs) (in 0.5 μM TTX). In contrast to the findings in the hippocampus, DHPG caused a significant increase in the frequency of sEPSCs and mEPSCs in the LA neurons (*Figure 1d–g*). Thus, taken together, pharmacological activation of mGluR5 enhanced both intrinsic and synaptic excitability in LA principal neurons in vitro.

What are the behavioral consequences of these cellular changes induced by mGluR5 activation? To address this question, we combined a discriminative fear conditioning procedure with targeted in vivo infusion of saline or DHPG directly into the basolateral amygdala (BLA) of awake, behaving rats (*Figure 2a–b*). Following context habituation (Days 1 and 2), animals were first subjected to in vivo infusions of saline into the BLA. This manipulation had no effect on the animal's freezing response during habituation to two tones (Day 3, *Figure 2c*) that were subsequently used for discriminative auditory conditioning. Rats were trained to discriminate the two tones of different frequencies – one (CS⁺) was paired with a foot shock (US) and the other was not (CS⁻) (*Figure 2a*). Using this training protocol, rats were conditioned to a relatively low intensity of foot shock (US: 0.4 mA, Day 3) that did not lead to any significant increase in the freezing response to the CS⁺ compared with the CS⁻, or tone habituation, 24 hr later (Testing, Day 4, *Figure 2c*). Thus, differential conditioning with a weak US by itself was unable to produce any detectable change in cue-specific fear (CS⁺-induced freezing). Next, the same animals received in vivo infusions of DHPG into the BLA, followed by the same sequence of tone habituation and weak-US conditioning (Day 4). In contrast to saline, infusions of DHPG in the same animals caused a significant increase in freezing to both the CS⁺ and CS⁻ during tone habituation (Day 4, *Figure 2c*). Moreover, the elevated levels of freezing elicited by the two tones were indistinguishable. This lack of discrimination was seen till the end of the conditioning session (*Figure 2—figure supplement 2c*).

These animals were then subjected to the same weak conditioning (Day 4). Surprisingly, despite DHPG triggering an immediate and significant enhancement in freezing to both the CS⁺ and CS⁻ (Day 4, *Figure 2c*), 24 hr later the rats were able to discriminate between the safe and dangerous tones, as evidenced by selectively increased freezing to the CS⁺ over CS⁻ (Testing, Day 5, *Figure 2c*). Thus, weak US conditioning, in conjunction with simultaneous in vivo activation of mGluR5 in the BLA, eventually caused a selective increase in cue-specific fear.

Is the failure to discriminate between the CS⁺ and CS⁻ immediately after the infusion of DHPG indicative of fear generalization? To address this question, we quantified freezing exhibited

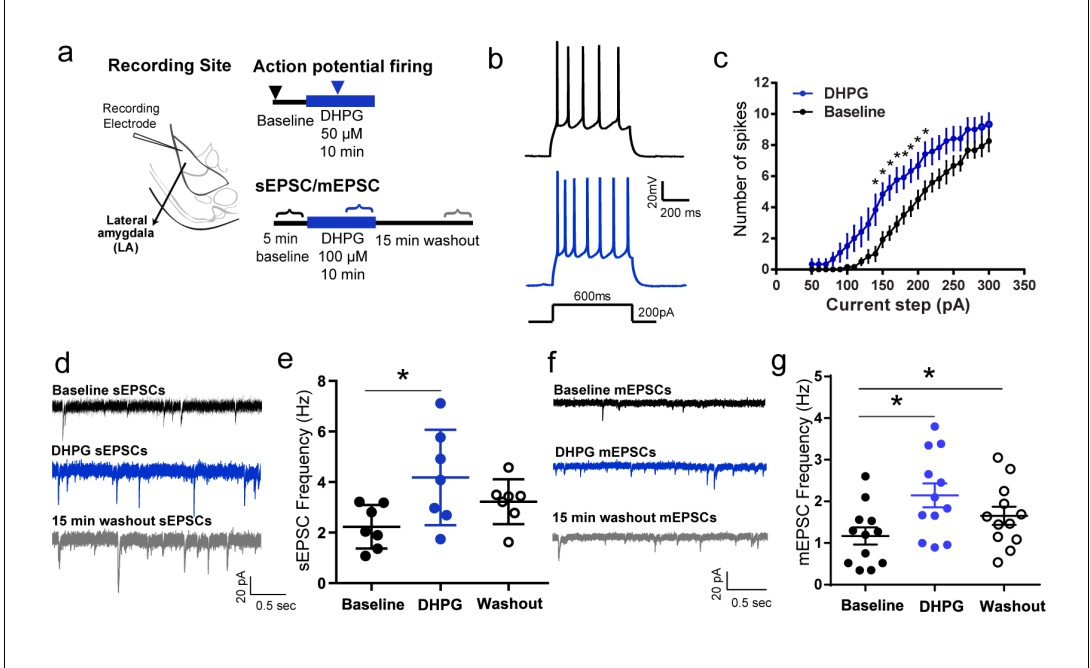

**Figure 1.** Pharmacological activation of mGluR5 using DHPG enhances intrinsic and synaptic excitability in LA principal neurons in vitro. (a) Placement of recording electrode in a coronal brain slice of the lateral amygdala (LA) (*left*). Schematic of experimental protocols (*right*). Current-clamp recordings of action potential firing (*right, top*) was compared using a range of current injections (600 ms, 50–300 pA, 10 pA steps) between baseline firing before DHPG application (▼) and during a 10 min bath application of 50 µM DHPG ( ▼ ). Voltage-clamp recordings of spontaneous EPSCs and miniature EPSCs were made (*right, bottom*) before ( ⌢ ), during ( ⌢ ) and 15 min after ( ⌢ ) bath application of 100 µM DHPG for 5 min each in the same neuron. (b) Representative traces of accommodating action potential firing from a LA principal neuron in response to depolarizing current injections (200 pA) before (*top*, black) and after DHPG application (*bottom*, blue). (c) Averaged number of spikes fired shows an increase in excitability after DHPG application across a range of current injections (n = 6 neurons, p<0.05). (d) Representative traces showing voltage clamp recordings ($V_{HOLD}$ = −70 mV) of sEPSCs from LA neurons before (*black*), during (*blue*) and 15 min after washout of DHPG (*grey*). (e) DHPG causes a significant increase in the mean frequency of sEPSCs that returns to baseline levels after washout (n = 7 neurons, p<0.05). (f) Representative traces showing voltage clamp recordings ($V_{HOLD}$ = −70mV) of mEPSCs from LA neurons before (*black*), during (*blue*) and 15 min after washout of DHPG (*grey*). (g) DHPG causes a significant increase in the mean frequency of mEPSCs that remains elevated even after 15 min of washout (n = 12 neurons, p<0.05). *p<0.05 in all the graphs.

The following source data is available for figure 1:

**Source data 1.** Data for individual cells representing the number of spikes generated in response to current injection (*Figure 1c*), frequency of sEPSCs (*Figure 1e*) and frequency of mEPSCs (*Figure 1g*) during baseline recordings, in the presence of DHPG and after washout of DHPG.

immediately before the onset of any tones. Interestingly, these animals exhibited high levels of freezing (33.6 ± 10.7%) even before the onset of the first tone (Pretone, measured as freezing levels 10 s immediately preceding the CS; *Figure 2d*). Moreover, these pretone freezing levels were not significantly different compared to tone-induced freezing (p=0.35, one-way repeated measures ANOVA). Further, freezing levels in the habituation context for a longer duration (1 min) preceding the tone were also high due to DHPG infusion (36.2 ± 14.1%), suggesting significant enhancement in context generalization (Day 4, *Figure 2d*). In other words, DHPG caused the animals to exhibit indiscriminate fear irrespective of whether the tones were present or not. However, 24 hr later these animals did not exhibit enhanced pretone freezing in the testing context (Testing, Day 5, *Figure 2d*), but only a selective increase in freezing to the CS$^{+}$ (Testing, Day 5, *Figure 2c*). Together, these behavioral results suggest that although DHPG activates the amygdala to produce indiscriminate fear of both tone and context, this state does not interfere with its ability to learn proper tone-shock associations that lead to a selective enhancement in cue-specific fear.

Finally, we confirmed that these behavioral effects were indeed due to DHPG, and not due to reconditioning on Day 4. A control group underwent the same behavioral protocol but received intra-BLA infusion of saline on Day 4 (*Figure 2—figure supplement 1*). To this end, these control animals

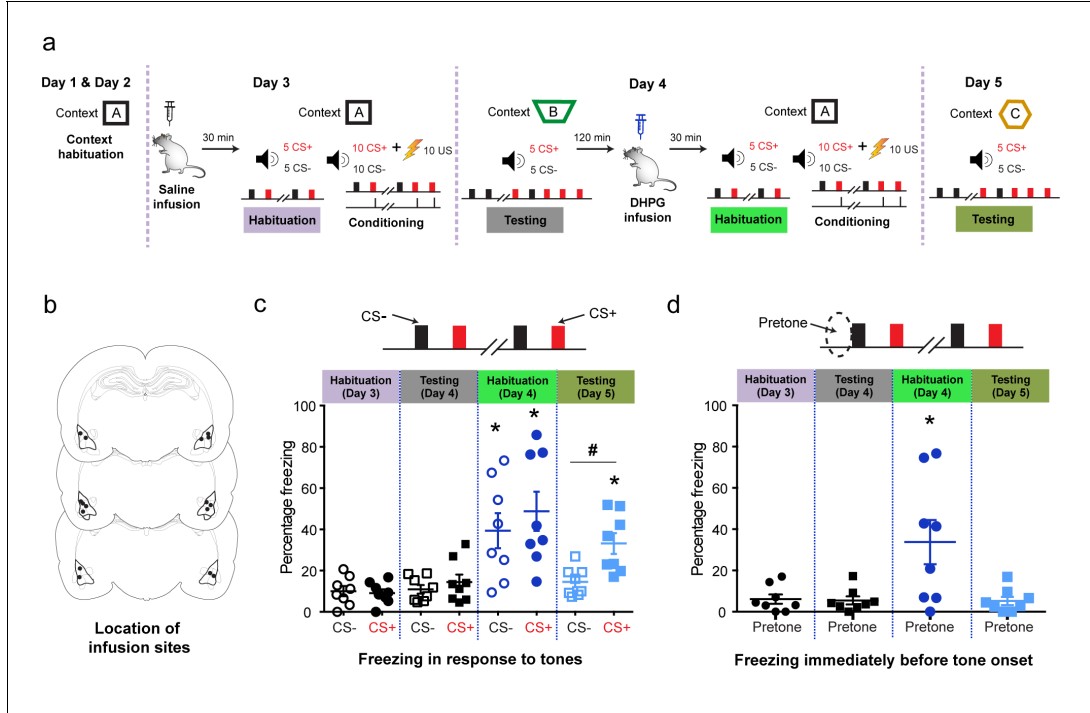

**Figure 2.** Targeted in vivo activation of mGluR5 in the BLA by itself initially causes indiscriminate fear, but it eventually leads to selective strengthening of cue-specific fear when combined with conditioning. (a) Experimental protocol for discriminative auditory fear conditioning (10 CS[+]-US pairings were interleaved with 10 CS[−] presentations during conditioning) using a weak US (0.4 mA) combined with in vivo infusion (1.0 µl per side) of saline (0.9% NaCl) followed by DHPG (50 µM of DHPG) into the BLA of the same animal. (b) Schematic coronal sections depicting cannula infusion sites for saline/DHPG. (c) Mean freezing levels 30 min after saline infusions on Day 3 showed no difference between the CS[+] and CS[−] during habituation (N = 8 rats, p>*0.05*). During testing 1 d after weak conditioning (Day 4), the CS[+] did not evoke higher freezing relative to either the CS[−] (p>0.05), or habituation (p>0.05). However, 30 min after DHPG infusion, the same animals exhibited significantly higher freezing to both CS[+] and CS[−] (*p<0.05), and the freezing levels were indistinguishable between CS[+] and CS[−] (Habituation, Day 4: p>0.05). Strikingly, subsequent conditioning in the presence of DHPG using the same weak US (Day 4, 0.4 mA) strengthened cue-specific fear. Thus, during testing 1 d after weak conditioning, the CS[+] evoked higher freezing relative to the CS[−] (Testing, Day 5: #, p<0.05), as well as CS[+]-evoked freezing 1 d after weak conditioning in saline (Testing, Day 4: *, p<0.05). (d) Mean freezing levels in the different spatial contexts immediately before the presentation of tones (pretone) during the same tests described in (c). Enhanced pretone freezing in the context was observed only immediately after DHPG infusion (Habituation, Day 4: *p<0.05). *p<0.05 between sessions in all graphs; #p<*0.05* between CS[+] and CS[−] within sessions in all graphs.

The following source data and figure supplements are available for figure 2:

**Source data 1.** Data for individual animals representing freezing response to CS+, CS- and pretone during the different phases of behaviour (***Figure 2c and d***).

**Figure supplement 1.** Control experiments using the same discriminative auditory fear conditioning with a weak US combined with in vivo infusion (1.0 µl per side) of saline (0.9% NaCl) followed by a second episode of the same saline infusion into the BLA of the same animal did not cause the increase in indiscriminate fear and cue-specific fear seen with DHPG infusion (as reported in ***Figure 2***).

**Figure supplement 1—source data 1.** Data for individual animals representing freezing response to CS+, CS- and pretone during the different phases of behaviour (***Figure 2—figure supplement 1b and c***).

**Figure supplement 2.** Additional analysis of freezing behavior during differential conditioning after DHPG infusion on Day 4.

**Figure supplement 2—source data 1.** Data for individual animals representing freezing response to CS+, CS- and pretone during the first and last trial of conditioning after DHPG infusion on day 4 (***Figure 2—figure supplement 2c***).

received the saline infusions twice (on Day 3 and Day 4) instead of saline (Day 3) followed by DHPG (Day 4). These animals did not exhibit any of the enhancing effects on indiscriminate and cue-specific fear seen with DHPG infusions.

Next, guided by these results, we investigated DHPG-induced modulation of plasticity mechanisms in LA slices that may underlie these distinct effects on specific versus indiscriminate fear. Accumulating evidence suggests that associative LTP in the LA serves as a synaptic mechanism for encoding memories of the CS–US association during fear conditioning (*Rogan et al., 1997*). Specifically, it has been proposed that the temporal overlap of CS-evoked EPSPs at thalamic input synapses onto LA neurons and US-induced depolarization of these same neurons leads to associative LTP, which serves as a cellular analogue of fear conditioning (*Blair et al., 2001*). Since we used a low intensity US that was not strong enough to cause robust fear conditioning, we adapted a weak associative pairing protocol, wherein EPSPs evoked by presynaptic stimulation of thalamic afferents were paired with a weak postsynaptic depolarization of LA neurons that is not effective in triggering synaptic potentiation (*Figure 3a–b*) (*Bauer et al., 2001*). Strikingly, the same weak pairing protocol,

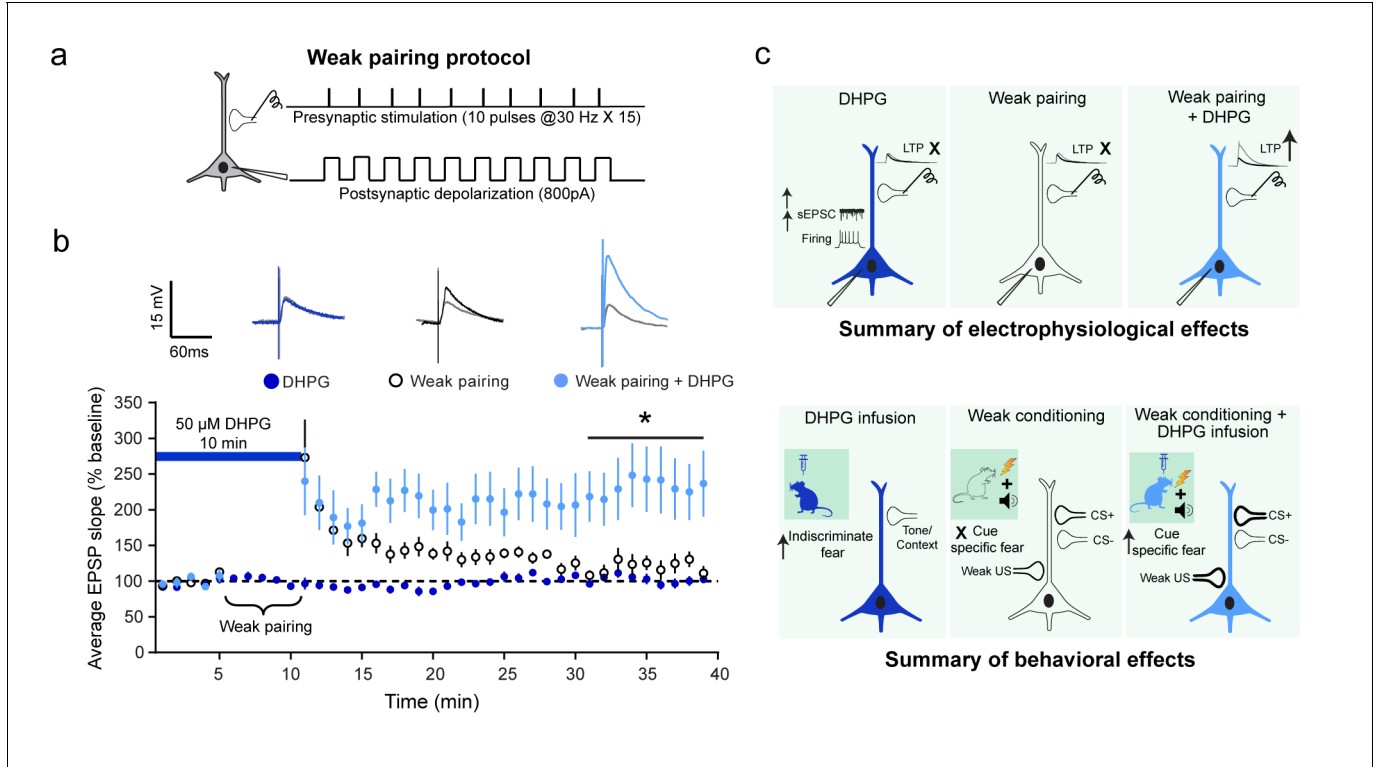

**Figure 3.** Although DHPG and a weak associative pairing protocol on their own did not elicit LTP, the two together caused a robust facilitation of LTP. (a) Schematic of the weak associative pairing protocol that involved pairing EPSPs evoked by presynaptic stimulation of thalamic afferents (trains of 10 pulses at 30 Hz, repeated 15 times) with weak postsynaptic depolarization of LA neurons (somatic injection of 800 pA current) timed to coincide with the peak of EPSPs evoked by presynaptic stimuli (see Materials and methods). (b) 10 min bath application of 50 µM DHPG (*blue bar*) alone did not induce LTP (●, n = 11 neurons, p>0.05). The weak pairing protocol (⌣) caused only a transient enhancement in the EPSP slope, but no LTP (○, n = 11 neurons, p>0.05). However, in the presence of DHPG, the same weak pairing protocol induced robust LTP (●, n = 12 neurons, *p<0.05). Superimposed representative EPSP traces before (*grey*) and 30 minutes after the three manipulations – DHPG alone (*dark blue*), weak pairing alone (*black*), and weak pairing + DHPG (*light blue*). (c) Summary of the electrophysiological effects of in vitro activation of mGluR5 in LA slices and how they relate to the behavioral effects of in vivo activation of mGluR5, as described in the text.

The following source data is available for figure 3:

**Source data 1.** Data for individual cells representing the EPSP slope, normalized to baseline with DHPG treatment, weak pairing and weak pairing in the presence of DHPG (*Figure 3b*).

when delivered in the presence of DHPG, triggered significantly larger LTP. However, DHPG treatment by itself did not induce LTP at thalamic inputs (*Figure 3b*).

## Discussion

Although earlier studies using pharmacological blockers implicate mGluR5 activity in the amygdala in both fear and anxiety, these did not offer a way of dissecting the two amygdala-dependent behaviors. The findings reported here provide new insights into how activation of the same mGluR5 receptor can specifically modulate intrinsic excitability and synaptic plasticity in neurons of the basolateral amygdala (BLA), thereby leading to distinct effects on specific versus indiscriminate fear. DHPG treatment alone enhanced LA neuronal excitability without eliciting any LTP in vitro (*Figure 3c*). Consistent with this, in vivo activation of mGluR5 caused an indiscriminate and immediate increase in freezing to both context and tones, without affecting cue-specific fear. The weak associative pairing protocol that did not elicit LTP reflects the association of the $CS^+$ with a weak US that failed to form a strong fear memory. Finally, although DHPG and the weak associative pairing on their own had no significant impact on LTP, the two together caused a robust facilitation of LTP, mirroring the significantly stronger fear memory formed by the same weak conditioning in the presence of DHPG in vivo (*Figure 3c*). In other words, though the synaptic plasticity triggered by CS-US association took place in the backdrop of heightened excitability through enhanced action potential firing and enhanced spontaneous synaptic activity in the BLA, it did not interfere with the effective encoding of cue-specific fear memory later on. This is in agreement with an earlier study that reported that injections of DHGP into the BLA before fear conditioning enhanced subsequent freezing to both context and the auditory cue paired with a weak footshock (*Rudy and Matus-Amat, 2009*). While this earlier study reported an enhancement in learned fear 24 hr after DHPG infusions, we find that this effect is not accompanied by generalization of fear. However, DHPG infusion causes an immediate increase in indiscriminate fear of both context and tones, a manifestation of anxiety-like behavior (*Duvarci et al., 2009*). Thus, unlike earlier studies, our findings demonstrate a distinct temporal separation between specific and indiscriminate fear triggered by targeted activation of mGluR5 in the BLA.

Comparison of our results with those reported by Rudy and Matus-Amat also highlights another key feature of the effects of DHPG in the BLA. In their study, the CS-US pairing was performed in the presence of DHPG but *without* any pre-exposure to tones during a habituation session – and fear memory was still enhanced. This suggests that it is the tone-shock pairing in the presence of DHPG that boosts the subsequent cue-specific fear learning, not the pre-exposure to tones. In other words, DHPG may persistently increase excitability of amygdalar neurons such that pairing $CS^+$ with even a weak US in the presence of DHPG is sufficient to subsequently enhance fear learning. Interestingly, a similar effect of DHPG on neuronal excitability has been observed in the hippocampus (*Cohen and Abraham, 1996*; *Ireland and Abraham, 2002*), though its behavioral consequences are not known. Future studies will be needed to examine if these cellular effects of DHPG in the hippocampus and amygdala also lead to enhancement of contextual fear learning. mGluR5 activation also contributes to the converse phenomenon – fear extinction – by increasing burst firing of infralimbic neurons in the medial prefrontal cortex (*Fontanez-Nuin et al., 2011*), which is similar to the enhanced firing of LA neurons reported here.

While animal models of anxiety-like behaviors have traditionally used assays such as the elevated plus-maze, open field tests, etc. we have employed a discriminative fear conditioning paradigm that allowed us to examine the effects of mGluR5 activation on both specific and non-specific fear within the same experimental framework (*Botta et al., 2015*; *Duvarci et al., 2009*). This enabled us to demonstrate that DHPG eventually led to a strengthening of cue-specific, but not generalized fear. However, the same manipulation caused an immediate increase in freezing to both tones, similar to what is seen in generalization of fear. Interestingly, closer inspection of this behavior revealed that the animals actually exhibited high freezing even before the onset of the $CS^+/CS^-$, which was indicative of abnormally high levels of non-specific fear of a novel spatial context irrespective of whether the tones were on or off (*Figure 2d*). Together, these short-term behavioral effects of DHPG infusion into the BLA are likely to represent high levels of anxiety-like behavior. Recent work points to a role for various amygdalar nuclei in a continuum of anxious/fearful behaviors spanning fear generalization, contextual freezing, and anxiety-like behavior on the elevated plus-maze (*Duvarci et al., 2009*;

*Ehrlich et al., 2009*). Moreover, while the present study explored mGluR5-dependent mechanisms in the BLA, there is also accumulating evidence for a pivotal role for amygdalar GABA$_A$ receptors in the modulation of these behaviors (*Botta et al., 2015*; *Ehrlich et al., 2009*).

We also report that DHPG increases the frequency of spontaneous synaptic events in LA neurons, which is in contrast to its reduction reported earlier in hippocampal area CA1 (*Fitzjohn et al., 2001*; *Snyder et al., 2001*). Moreover, although DHPG by itself failed to elicit any lasting changes in synaptic efficacy in the LA, it has been shown to elicit LTD in area CA1 (*Huber et al., 2000*). This contrast is also seen in aberrant mGluR-plasticity in a mouse model of Fragile X Syndrome (FXS) – hippocampal mGluR5-LTD is enhanced while amygdalar mGluR5-LTP is impaired (*Huber et al., 2002*; *Suvrathan et al., 2010*). These findings highlight the need for investigating the molecular basis of the divergent effects of mGluR5 activation in the amygdala versus hippocampus. Finally, our findings may also be relevant to the paradoxical results from a functional MRI study reporting an *inverse* relationship between fear-specific amygdala activation and anxiety scores in FXS individuals (*Kim et al., 2014*). While pharmacological antagonism of amygdalar mGluR5 prevents both anxiety and fear, our results provide insights into how activation of the same receptor can separately modulate specific versus indiscriminate fear without affecting both simultaneously, thereby providing a distinction between the two. Thus, these findings offer a new framework, spanning multiple levels of neural organization, for analyzing mGluR-plasticity in the amygdala, and its implications for emotional dysfunction.

## Materials and methods

### Experimental animals

Naïve 5–7 weeks old adolescent male Sprague-Dawley rats (RRID: RGD_734476) weighing 200–300 grams (*Kim et al., 2007*; *Miracle et al., 2006*) and housed (Animal care Facility, National Centre for Biological Sciences, Bangalore, India) in groups of two were used in the study. They were maintained on a 14 hr/10 hr light/dark cycle and had access to water and a standard diet *ad libitum*. The age of animals was selected to match earlier studies with mGluR activation. The same age group was maintained for behavior experiments to make the interpretation of the results comprehensive. All experiments were conducted in accordance with the guidelines of the CPCSEA, Government of India and approved by the Institutional Animal Ethics Committee of National Centre for Biological Sciences.

### Slice preparation

Rats were anaesthetized using halothane and decapitated. The brain was quickly dissected out and transferred to oxygenated, ice-cold artificial cerebrospinal fluid (ACSF) containing (in mM): 115 NaCl, 25 glucose, 25.5 NaHCO$_3$, 1.05 NaH$_2$PO$_4$, 3.3 KCl, 2 CaCl$_2$ and 1 MgSO$_4$. Whole-brain coronal slices (400 μm) were obtained using a Vibratome (VT1000S, Leica Biosystems, Wetzlar, Germany), transferred to a submerged holding chamber containing oxygenated ACSF and allowed to recover for 1 hr at room temperature. Individual slices were then transferred to a submerged recording chamber (28 ± 2°C) and neurons were visually identified using an upright differential interference contrast microscope (BX50WI, Olympus, Tokyo, Japan).

### Whole-Cell recordings

Patch pipettes (3–5 MΩ resistance, ~2 μm tip diameter) pulled from thick-walled Borosilicate glass using a P1000 Flaming/Brown micropipette puller (Sutter Instruments, Novato, California, USA) and filled with an internal solution containing (in mM) 120 K-gluconate, 20 KCl, 10 HEPES, 4 NaCl, 4 MgATP, 0.3 NaGTP, 0.2 EGTA and 10 phosphocreatine (pH 7.4, ~285 mOsm) were used to patch on to principal neurons in the Lateral Amygdala. For voltage clamp experiments, potassium was replaced with equimolar cesium in the internal solution. In experiments where evoked responses were recorded, a bipolar electrode (25 μm diameter Platinum/Iridium, FHC) connected to an ISO-Flex stimulus isolator (A.M.P.I.) was used to stimulate the thalamic inputs to the amygdala. Data were recorded with an EPC-9 amplifier (HEKA Elektronik, Lambrecht, Germany), filtered at 2.9 kHz and digitized at 20 kHz. Data acquisition and stimulus delivery were performed using Patchmaster software (HEKA Elektronik, Lambrecht, Germany), while analysis of electrophysiological data was performed using custom-written programs in IGOR Pro (Wavemetrics Inc., Nimbus, Portland, USA,

RRID: SCR_000325), unless otherwise stated. Neurons were used for recording if the initial resting membrane potential ($V_m$) $\leq$-60 mV, series resistance ($R_s$) was 15–25 MΩ and were rejected if the $R_s$ changed by >20% of its initial value. For all recordings, neurons were held at −70 mV.

## sEPSC/mEPSC recordings

LA neurons were clamped at −70 mV and spontaneous excitatory postsynaptic currents (sEPSCs) were isolated with picrotoxin (75 µM). Miniature excitatory postsynaptic currents were isolated by clamping LA neurons at −70 mV in the presence of TTX (0.5 µM) and picrotoxin (75 µM). For both sets of experiments baseline current traces of 5 min duration (recorded at least 5 min after achieving whole-cell configuration) were recorded. 100 µM of 3,5-Dihydroxyphenylglycine (DHPG, Abcam, Cambridge, UK) was applied in the bath solution for 10 min and then washed out for 15 min (*Figure 1a*). Continuous current traces of 5 min duration were analyzed immediately before DHPG application. During DHPG application the last 5 min of traces were analyzed and after washout 5 min traces were analyzed starting 10 min from the onset of DHPG washout. Mini Analysis Program (Synaptosoft Inc., Fort Lee, New Jersey, USA, RRID: SCR_002184) was used for analyzing the traces. A total of 7 cells (from 5 animals) were used for the sEPSC experiments. 12 cells (from 7 animals) were used for the mEPSC experiments.

## Frequency-current relationship

To obtain the frequency-current (*f-I*) relationship, the number of action potentials fired during a 600 msec pulse of depolarizing current injections from 50 pA to 300 pA was recorded. Two pulses for each current magnitude were recorded and the average number of spikes was obtained for each current magnitude. After measuring baseline firing, the firing properties of the same neuron were measured during a 10 min bath application of 50 µM DHPG, starting 5 min from the onset (*Figure 1a*). A total of 6 cells from three animals were used for this experiment.

## Long-term potentiation (LTP) experiments

Evoked responses of neurons in the lateral amygdala (LA) were recorded in response to stimulation of internal capsule fibers containing thalamic inputs to the LA (at 0.05 Hz). In the 'DHPG' group, 50 µM DHPG was bath applied to the slices for 10 min after achieving a stable whole-cell configuration followed by a 30 min washout. In the 'Weak Pairing' group, potentiation was induced at the thalamic inputs after recording a stable baseline for 5 min. LTP induction consisted of 10 stimuli delivered at 30 Hz and repeated 15 times with an interval of 30 s between trains (*Figure 3b*). The postsynaptic neuron was simultaneously depolarized at the peak of the EPSP with a current injection of 800 pA for a period of 5 msec (experimental protocol adapted from [*Bauer et al., 2001*]). In the 'Weak Pairing + DHPG' group, LTP was induced as described above in the presence of 50 µM DHPG in the bath solution. The amplitudes of EPSPs during baseline acquisition were maintained within 5–10 mV and neurons were excluded from the analysis if the LTP induction protocol was not carried out within 15 min after achieving whole-cell configuration. LTP was quantified in Igor Pro (Wave Metrics Inc.) by measuring the initial slope of the EPSP, calculated during a 1–2 ms period, by placing cursors within the 10 to 90 range of the baseline EPSP slope. The same cursor settings were maintained for slope measurements over the entire pre and post-LTP time course for each cell. The measured EPSP slopes were then normalized to the average baseline value for each cell. A total of 11 to 12 cells from 5 to 8 animals were used for each group in this experiment.

## Experimental paradigm for behavioral experiments

The animals were handled for 2 days to habituate to the experimenter before the start of any experimental procedures. After this 2 day handling period, the animals were bilaterally implanted with stainless steel cannulae targeted at the amygdala. After the surgery, the animals were allowed to recover for 7–8 days. Then, the animals were habituated to the conditioning context on Day 1 and Day 2 for 25 min per day (*Figure 2a*, left). On Day 3, the animals were subjected to targeted infusion of saline into the LA. After a delay of 30 min post-infusion, the animals underwent a tone habituation session followed immediately by fear conditioning (*Figure 2a*, second from left). On Day 4, first the animals underwent fear recall session. Then, the animals were returned to the homecage for two hours. After two hours, the same animals were subjected to DHPG infusion bilaterally into the BLA.

Then, following a 30 min interval the animals underwent tone habituation and subsequent fear conditioning (*Figure 2a*, second from right). Finally, on Day 5, the animals were subjected to fear recall again (*Figure 2a*, right). To rule out the effect of fear conditioning the same animal twice and to understand the effect specific to mGluR activation, another group of animals was subjected to the same protocol with the exception of infusion of saline on Day4 instead of DHPG (*Figure 2—figure supplement 1a*). After the conclusion of the experiment, the rats were anesthetized and their brains were collected to confirm the anatomical location of the infusion site. All the behavioral experiments were performed in the light cycle between 9.00 am and 2.00 pm.

## Surgical procedure

For targeted infusion of DHPG into the BLA, rats were surgically implanted with bilateral, chronic, intracranial stainless steel guide-cannulae (7 mm long, 24 gauge, Plastic One, Roanoke, Virginia, USA) aimed at the dorsal half of the BLA.

Rats were subjected to anesthesia with 5% isoflurane (Forane, Asecia Queensborough, UK) and then maintained under anesthesia with 1.5–2% isoflurane. The level of anesthesia was regularly monitored throughout the procedure using the pedal withdrawal reflex to toe pinch. The animal was placed and head fixed on a stereotaxic frame. Body temperature of rats was maintained with a heating pad. Burr holes were drilled at the stereotaxic coordinates of the BLA (2.8 mm posterior to bregma and ±5.2 mm lateral to midline, (*Paxinos and Watson, 2009*). Stainless steel guide cannulae were then implanted using the stereotaxic frame (7.0 mm ventral from the brain surface(*Paxinos and Watson, 2009*). The implant was secured using anchor screws and dental acrylic cement for further infusion experiments. Dummy cannulae (28 gauge) with 0.5 mm projection were inserted into the cannulae to prevent clogging. Rats were allowed to recover for 7–8 days following surgery. In the post-surgery period the animals were singly housed in separate cages.

A total of 17 animals were implanted with stainless steel guide-cannulae. Two animals were excluded from the study because the positioning of the tip of both the cannulae was not in the BLA. Eight animals were used for the main experimental group (saline infusion on Day 3 and DHPG infusion on Day 4) and seven animals were used in the control group (Saline infusion on both Day 3 and Day 4). The locations of the cannulae placement for the eight animals used in the experimental group study are shown in *Figure 2b*.

## Pharmacological infusion of DHPG into the basolateral amygdala

Intra-amygdala infusion of DHPG was performed using standard pressure injection methods (*Ghosh and Chattarji, 2015*). During the infusion procedure, the rats were kept in their home cages and injection cannulae with 1 mm projection (28 gauge, Plastic One, Roanoke, Virginia, USA) were inserted through the guide-cannulae. The injection cannula was connected to a Hamilton syringe (10 μl) using a polyethylene tubing (Plastic One, Roanoke, Virginia, USA). The Hamilton syringe was mounted on an infusion pump (Harvard Apparatus, Holliston, Massachusetts, USA). Rats were infused bilaterally with one hemisphere at a time. Either vehicle (0.9% (vol/vol) NaCl, 1.0 μl per side [*Akirav et al., 2006*; *Rudy and Matus-Amat, 2009*]) or DHPG (1.0 μl per side, 50 mM in saline; Abcam, Cambridge, UK) was infused at a rate of 0.2 μl min$^{-1}$. The injection cannula was taken out 5 min after the end of infusion, to allow the drug to diffuse into the tissue. After the completion of experiments, cannulae placement was confirmed using standard histological methods (*Figure 2b*).

## Fear conditioning and recall protocol

Fear conditioning and fear memory recall tests took place in different contexts placed inside sound-isolation boxes (Coulbourn Instruments, Whitehall, Pennsylvania, USA). Conditioning was performed in a box with metal grids on the floor (context A: 30 centimeters wide × 25 cm deep × 30 cm high, no odor). Fear recall was performed in two contexts, one was a modified homecage (context B: 35 centimeters wide 20 centimeters deep × 40 cm high, mint odor), and the second was a hexagonal box made of plexiglass (context C: 30 cm wide × 30 cm deep ×30 cm high, 0.1% acetic acid odour). Lighting conditions and floors and walls were different between the three contexts. All chambers were cleaned with 70% alcohol before and after each experiment.

The behavior of the animals was recorded using a video camera mounted on the wall of the sound isolation box and a frame grabber (sampling at 30 Hz). The videos were analyzed offline for

further quantification of freezing behavior (*Ghosh and Chattarji, 2015*; *Suvrathan et al., 2014*). Infrared LED cues were placed on the walls of the experimental chambers. These cues were activated in coincidence with auditory stimuli to monitor the tone-evoked freezing response offline. A programmable tone generator and shocker (Habitest system, Coulbourn Instruments, Whitehall, Pennsylvania, USA) were used to deliver tones and foot-shock during the experiment. Foot-shocks were delivered through the metal grids on the floor of the conditioning chamber. The tone was played using a speaker (4 Ω, Coulbourn Instruments, Whitehall, Pennsylvania, USA) placed inside the experimental chamber.

During context habituation, the animals were allowed to explore context A for 25 min in each session. Next, in the tone habituation session (*Figure 2a*) the animals received five presentations of two tones each in random order (total duration of 10 s, consisting of either 15 kHz tone presented as clicks at 1 Hz frequency or continuous tones at 5 kHz; 5 ms rise and fall, 70 ± 5 dB sound pressure level) in context A. This was immediately followed by a differential fear conditioning protocol, wherein one of these tones (CS$^+$: 5 kHz continuous tone) was paired (ten pairings, average inter-trial interval <ITI> = 70 s, with a range of 40–100 s) with a co-terminating 1 s scrambled foot shock (US: 0.4 mA, 1 s), whereas the other was not (CS$^-$: 15 kHz clicks). Thus, ten CS$^+$-US pairings with ten randomly interleaved CS$^-$ presentations were given in random order during this differential conditioning procedure. In the testing session, the animals were introduced into either context B or context C. The animals were allowed to explore the context for 3 min and then presented with both CS$^-$ and CS$^+$ (10 s, five trials each, <ITI> = 70 s, 40–100 s) in a pseudo-random order (3 CS$^-$ followed by two alternate CS$^+$ and CS$^-$ followed by 3 CS$^+$) to test recall of fear memory.

## Behavioral analysis

Behavioral response was hand scored offline using video recordings of tone habituation and all testing sessions. Response to the auditory stimuli was evaluated in the form of a freezing response. Freezing was defined as the absence of movement except due to respiration (*Blanchard and Blanchard, 1988*). The time spent freezing during the presentation of the tone was converted into a percentage score (*Figure 2c*). The percentage freezing level was measured in every context/session for 10 s immediately before the presentation of the first tone trial to assess pretone freezing in absence of any auditory stimulus. This was defined as the freezing in the pretone period (*Figure 2d*). Percentage of freezing was also measured for one minute prior to the presentation of the first tone trial to assess context-dependent freezing for a sustained period. This quantification across groups was done blindly.

## Histology

After the experiment was concluded, rats were deeply anesthetized (ketamine/xylazine, 100/20 mg per kg). Coated silver wires with a bare tip were inserted into the cannulae (1 mm projection) and electrolytic lesions (20 µA, 20 s) were made to mark the in vivo infusion sites. The animals were then perfused transcardially with ice-cold saline (0.9%) followed by 10% (vol/vol) formalin. The perfused brain was left in 10% (vol/vol) formalin overnight. Coronal sections (80 µm) were prepared using a vibratome (VT 1200S, Leica Microsystems, Wetzlar, Germany) and mounted on gelatin-coated glass slides. Sections were stained with 0.2% (wt/vol) cresyl violet solution and mounted with DPX (Sigma-Aldrich, St. Louis, Missouri, United States). The slides were imaged to identify and reconstruct infusion sites (*Figure 2b*).

## Statistical analysis

All values are expressed as mean ± SEM. Each data set was evaluated for statistical outliers, which were defined as greater than twice the standard deviation away from the mean. According to this criterion, one neuron was excluded from mEPSC experiment in *Figure 1g* and one neuron was excluded from the 'Weak pairing + DHPG' group in *Figure 3b*. The number of spikes analyzed for the comparisons of frequency-current relationships is a discreet variable. Therefore, paired Kolmogorov–Smirnov test was performed to compare before and during DHPG conditions. One-way repeated measures analysis of variance (ANOVA) followed by post hoc Sidak's test was used for the comparisons of sEPSC and mEPSC frequencies before, during DHPG and 15 min after washout. The sEPSC and mEPSC datasets were found to have normal distributions for all the three conditions,

with no statistical difference in the variances. Freezing analysis was performed to evaluate the learning induced changes in behavioral responses. Conditioning induced changes in freezing to $CS^-$ and $CS^+$ were analyzed by repeated measures ANOVA as all the data passed the normality test. Further, post hoc Sidak's test was used to analyze the differential freezing responses in different sessions and the differential freezing responses to $CS^-$ or $CS^+$. The variances for all the datasets of $CS^-$ and $CS^+$ were statistically similar except for the habituation session on Day 4. Repeated measures ANOVA was also used to analyze the pretone freezing across contexts/sessions followed by post hoc Sidak's test where all the datasets were normal. The variance of pretone freezing during the habituation session on Day 4 was more than the other sessions. Two-way repeated measures ANOVA was used to analyze the percentage EPSP slopes across groups in the LTP experiments. This was followed by Tukey's post-hoc test for comparison between groups for each minute. All the data sets passed the normality test. All statistical tests were performed using GraphPad Prism (GraphPad software Inc., La Jolla, California, USA, RRID: SCR_002798).

## Acknowledgements

This work was supported by the Department of Atomic Energy and Department of Biotechnology, Government of India, the Wadhwani Foundation, and the Madan & Usha Sethi Fellowship.

## Additional information

### Funding

| Funder | Grant reference number | Author |
| --- | --- | --- |
| Department of Atomic Energy, Government of India | NCBS-4143 | Sumantra Chattarji |
| Department of Biotechnology, Ministry of Science and Technology | DBT-BT/MB-CNDS/2013 | Sumantra Chattarji |
| Wadhwani Foundation | | Sumantra Chattarji |
| Madan and Usha Sethi Fellowship | | Sumantra Chattarji |

The funders had no role in study design, data collection and interpretation, or the decision to submit the work for publication.

### Author contributions

MMR, Conceptualization, Data curation, Formal analysis, Validation, Investigation, Methodology, Writing—original draft, Writing—review and editing, Performed the in vivo pharmacological and behavioral experiments and related analyses; SK, Conceptualization, Data curation, Formal analysis, Validation, Investigation, Methodology, Performed the slice electrophysiological experiments and analyses; GF, Conceptualization, Data curation, Formal analysis, Validation, Investigation, Writing—original draft, Writing—review and editing, Performed the slice electrophysiological experiments and analyses; SC, Conceptualization, Supervision, Funding acquisition, Methodology, Writing—original draft, Project administration, Writing—review and editing

### Author ORCIDs

Mohammed Mostafizur Rahman, http://orcid.org/0000-0002-9355-4867
Sumantra Chattarji, http://orcid.org/0000-0001-9962-3635

### Ethics

Animal experimentation: All animal care and experimentation procedures were approved by the Institutional Animal Ethics Committee, National Centre for Biological Sciences (Approval No: SC-5/2009) and Committee for the Purpose of Control and Supervision of Experiments on Animals, Government of India (Registration No: 109/CPCSEA).

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
