## [Decision Letter]

Thank you for submitting your article "Activation of the same mGluR5 receptors in the amygdala causes divergent effects on specific versus indiscriminate fear" for consideration by *eLife*. Your article has been favorably evaluated by a Senior Editor and three reviewers, one of whom, Jennifer L Raymond (Reviewer #2), is a member of our Board of Reviewing Editors. The following individual involved in review of your submission has agreed to reveal their identity: Fabricio do Monte (Reviewer #1).

The reviewers have discussed the reviews with one another and the Reviewing Editor has drafted this decision to help you prepare a revised submission.

Summary:

The authors address the interesting question of the role of mGluR5 in fear learning and anxiety. The authors demonstrate that activation of mGluR5, with the agonist DHPG in the amygdala, promotes subsequent fear learning to a specific sound. Mechanistic insight into this phenomenon is provided by showing that coactivation of mGluR5 with a subthreshold LTP inducing stimulation leads to robust and persistent LTP, and that DHPG increases excitability of amygdala neurons, which likely underlies the enhancement of LTP. The authors also make a case that mGluR5 activation also leads to indiscriminate fear and anxiety by showing that DHPG injected in the amygdala during a "tone only" habituation period increases freezing behavior that is unrelated the tone. Perhaps most interesting is the observation that specific tone-shock associations can be learned in the presence of DHPG, even though the animals are exhibiting indiscriminate freezing in response to CS^+^, CS^-^ and the context on the day of training. The manuscript is well written, and the study was generally well designed.

Essential revisions:

The reviewers agreed that two key control experiments are necessary to support the authors' claims:

1) Most notably, in Figure 2, the authors concluded that intra-LA infusion of DHPG increased freezing to both CSs. However, freezing during the retrieval test (habituation, day 4) was compared with freezing levels observed before the conditioning (habituation, day 3) or in a completely neutral context (testing, context B). The increased freezing levels during the retrieval test might simply be the result of returning the animals to the context in which they were initially conditioned (context A). Similarly, the discrimination between CS^+^ and CS^-^ observed on day 5 may be caused exclusively by the extra-conditioning session that animals received on the previous day. Therefore, to confirm that the observed increase in freezing response was due to intra-LA infusion of DHPG, a control group undergoing the same behavioral protocol, but receiving intra-LA infusion of saline on day 4, needs to be performed. The lack of a saline-control group becomes more problematic in Figure 2, when the authors compare contextual freezing in the conditioned context (context A) versus contextual fear in two different safe contexts (context B and C).

2) The interpretation of the increase in sEPSCs in Figure 1 with DHPG in amygdala slices is ambiguous. Because the authors did not do this in TTX, the increase in sEPSCs is likely due to increases in spontaneous firing of other neurons in the slice. Additional experiments to determine the synaptic effects of DHPG are needed.

In addition, the reviewers raised several points that should be addressed with revisions to the text:

a) Rudy and Matus-Amat (2009) have demonstrated that infusion of DHPG into the basolateral amygdala increases freezing induced by a weak foot shock conditioning. This work should be acknowledged in the Introduction, and a clear explanation of how the present manuscript differs from the previous study needs to be provided. More generally, the authors should be sure to emphasize what is novel in their findings, given that it known that mGluR5 is required for fear memory and LTP in the amygdala, and DHPG can prime hippocampal neurons for LTP.

b) The subjects of the present study were 5-7 weeks old male Sprague-Dawley rats. Rats at this age are still considered adolescents. Considering that adolescent rats can differ from adults in learning and memory, as well as in fear responses, the authors should make it clear that they are using adolescent animals. In addition, they should cite previous studies on fear conditioning using adolescent rats in the Introduction and Discussion.

c) Based on the map drawings of histological placement presented on Figure 2, most of the amygdala infusions were located in the basolateral amygdala (which involves the lateral plus the basal nuclei), rather than in the lateral amygdala only. The authors should correct this throughout the manuscript. In addition, adding a representative picture of the site of infusion in the amygdala would be very informative for the readers.

d) The Abstract and the last sentence of the Discussion talk about a new "framework" for examining mGluR5-plasticity. What is that framework? Using agonist instead of antagonist? Thinking of fear and anxiety as independently modulated?

e) In the authors' schematic model they suggest that pairing of CS^+^ and DHPG during the habituation period is required for subsequent fear association with the CS^+^, but they do not actually test this. Does DHPG injected in the amygdala in their home cage without a habituation period or pre-exposure to CS^+^ enhance subsequent fear learning? DHPG may persistently increase excitability of amygdalar neurons (as occurs in hippocampus) such that presentation of CS^+^ with the weak US after DHPG is sufficient to enhance learning.

f) The authors also trained mice for fear learning to specific tones in different contexts. What is the role of the different contexts here? Does DHPG affect context specific fear learnings as well as to tones? This would strengthen the authors' conclusion that mGluR5 promotes discriminate learning.

g) The authors demonstrate that DHPG increases excitability of amygdalar neurons and so it is unclear if mGluR5 activity specifically is required to promote fear learning and anxiety or if simply increasing excitability will suffice. One way to test this would be to infuse a GABAA blocker into the amygdala during tone habituation, and test if this increases subsequent CS^+^ fear learning and LTP.

h) What is the behavior like during the conditioning session on Day 4, and especially towards the end of that session? Is there any evidence that the pretone freezing and the freezing to the CS^-^ start to go away when the associative conditioning begins?

---

## [Author Response]

*Essential revisions:*

*The reviewers agreed that two key control experiments are necessary to support the authors' claims:*

*1) Most notably, in Figure 2, the authors concluded that intra-LA infusion of DHPG increased freezing to both CSs. However, freezing during the retrieval test (habituation, day 4) was compared with freezing levels observed before the conditioning (habituation, day 3) or in a completely neutral context (testing, context B). The increased freezing levels during the retrieval test might simply be the result of returning the animals to the context in which they were initially conditioned (context A). Similarly, the discrimination between CS^+^ and CS^-^ observed on day 5 may be caused exclusively by the extra-conditioning session that animals received on the previous day. Therefore, to confirm that the observed increase in freezing response was due to intra-LA infusion of DHPG, a control group undergoing the same behavioral protocol, but receiving intra-LA infusion of saline on day 4, needs to be performed. The lack of a saline-control group becomes more problematic in Figure 2, when the authors compare contextual freezing in the conditioned context (context A) versus contextual fear in two different safe contexts (context B and C).*

We agree with the reviewers regarding the need for two key control experiments and we thank them for raising these issues. As summarized below, we have now performed both experiments and the new data are included in the revised manuscript.

1) As suggested, to confirm that the observed increase in the freezing response was due to intra-LA infusion of DHPG, a control group undergoing the same behavioral protocol but receiving intra-LA infusion of saline on Day 4, were tested. To this end we infused saline twice (on Day 3 and Day 4) instead of saline (Day 3) followed by DHPG (Day 4).

We found that during habituation on Day 4, the saline infused rats showed enhanced freezing, compared to habituation on Day 3, only to CS^+^ and not to CS^-^. Further, the animals showed cue-specific fear (significantly higher freezing in response to CS^+^ relative to CS^-^). Therefore, the indiscriminate fear observed in the DHPG-infusion group was not due to reintroduction to the same context where the animals were conditioned earlier, but a specific effect of DHPG infusion. Similarly, the saline-followed-by-saline infusion control group did not show enhancement of cue-specific fear on Day 5, thereby ruling out any effect of reconditioning. We also observed that the pretone freezing to the context is not enhanced in the saline-followed-by-saline group during habituation on Day 4, thereby confirming that the indiscriminate fear was caused by the DHPG infusion.

We have added a supplementary figure (Figure 2—figure supplement 1) depicting these new results and the related description has been added to the revised Results (fifth paragraph).

*2) The interpretation of the increase in sEPSCs in Figure 1 with DHPG in amygdala slices is ambiguous. Because the authors did not do this in TTX, the increase in sEPSCs is likely due to increases in spontaneous firing of other neurons in the slice. Additional experiments to determine the synaptic effects of DHPG are needed.*

We are grateful to the referee for raising this point. As advised, we have now carried out recordings of miniature excitatory postsynaptic currents (mEPSCs) in the presence of TTX. New data collected from these recordings show that even in the presence of 0.5µM TTX there is a significant increase in the frequency of mEPSCs elicited by bath application of 100µM DHPG, followed by a 15 washout (One-way repeated measures ANOVA followed by Holm-Sidak’s multiple comparison test). This clearly demonstrates the synaptic effects of DHPG, the point highlighted by the referee. We have reported this new result in the revised text and added the data to revised Figure 1. We have also added it to the Results section (first paragraph).

*In addition, the reviewers raised several points that should be addressed with revisions to the text:*

*a) Rudy and Matus-Amat (2009) have demonstrated that infusion of DHPG into the basolateral amygdala increases freezing induced by a weak foot shock conditioning. This work should be acknowledged in the Introduction, and a clear explanation of how the present manuscript differs from the previous study needs to be provided. More generally, the authors should be sure to emphasize what is novel in their findings, given that it known that mGluR5 is required for fear memory and LTP in the amygdala, and DHPG can prime hippocampal neurons for LTP.*

We thank the referee for raising these important points. We have made revisions in the relevant sections of both the Introduction and Discussion to include the specific points mentioned by the referee.

a) The revised Introduction now clearly acknowledges the earlier study by Rudy and Matus-Amat (Rudy and Matus-Amat, 2009), in which they showed that infusion of DHPG into the basolateral amygdala increases freezing induced by conditioning using a weak foot shock. Also, we have highlighted how the present study differs from the earlier study and how the strategy employed here attempts to address questions not examined by Rudy and Matus-Amat (Introduction, last paragraph).

b) Furthermore, we have also revised the Discussion to emphasize the novelty of our findings in light of earlier reports on mGluR5 and amygdalar LTP, fear memory and anxiety (Discussion, first paragraph). Finally, we now mention findings on DHPG priming of hippocampal LTP and how they compares with our results in the amygdala (Discussion, second paragraph), which is also related to points raised in part e) below.

Together, we have made extensive revisions in the Introduction and Discussion to emphasize the novelty of our findings, and how they compare/contrast with earlier reports (Introduction, last paragraph; Discussion, first and second paragraphs).

*b) The subjects of the present study were 5-7 weeks old male Sprague-Dawley rats. Rats at this age are still considered adolescents. Considering that adolescent rats can differ from adults in learning and memory, as well as in fear responses, the authors should make it clear that they are using adolescent animals. In addition, they should cite previous studies on fear conditioning using adolescent rats in the Introduction and Discussion.*

We have revised the manuscript to mention that adolescent rats were used (revised Methods subsection “Experimental Animals”). We have also added the relevant references as advised by the referee (Kim et al., 2007; Miracle et al., 2006).

*c) Based on the map drawings of histological placement presented on Figure 2, most of the amygdala infusions were located in the basolateral amygdala (which involves the lateral plus the basal nuclei), rather than in the lateral amygdala only. The authors should correct this throughout the manuscript. In addition, adding a representative picture of the site of infusion in the amygdala would be very informative for the readers.*

As suggested by the reviewer, we have made corrections throughout the manuscript to mention that the infusions were targeted at the basolateral amygdala (BLA) instead of the lateral amygdala. We have also added a representative image of the site of infusion in Figure 2—figure supplement 2. We thank the referee for this suggestion.

*d) The Abstract and the last sentence of the Discussion talk about a new "framework" for examining mGluR5-plasticity. What is that framework? Using agonist instead of antagonist? Thinking of fear and anxiety as independently modulated?*

We have revised the Discussion (first paragraph) to provide greater clarity on this important point raised by the referee. The findings reported here provide a framework for thinking about how fear and anxiety can be modulated independently by the same receptor in the BLA by using an mGluR5-agonist instead of an antagonist. Further, we find a similar separation at the cellular level between modulations of associative synaptic plasticity versus intrinsic plasticity. These represent a departure from earlier strategies using mGluR5-antagonists that did not explain how activation of the same receptor gives rise to both fear and anxiety. We have also made small modifications in the Abstract within the word limits allowed.

*e) In the authors' schematic model they suggest that pairing of CS^+^ and DHPG during the habituation period is required for subsequent fear association with the CS^+^, but they do not actually test this. Does DHPG injected in the amygdala in their home cage without a habituation period or pre-exposure to CS^+^ enhance subsequent fear learning? DHPG may persistently increase excitability of amygdalar neurons (as occurs in hippocampus) such that presentation of CS^+^ with the weak US after DHPG is sufficient to enhance learning.*

We apologize for the lack of clarity on this important point. Assuming that the “schematic model” the reviewer is referring to is Figure 3, this figure summarizes the behavioral results by making three points:

1) Presentation of the two tones (CS^+^ and CS^-^) caused an indiscriminate and immediate increase in freezing to both tones, as well as the context (left panel). However, subsequent fear association only enhanced freezing to the CS^+^, i.e. cue-specific fear, without increasing fear of the CS^-^.

2) In the absence of DHPG, the association of the CS^+^ with a weak US failed to form a strong fear memory (middle panel).

3) However, a significantly stronger fear memory (specific to the CS^+^, but not the CS^-^) was formed by the same weak conditioning in the presence of DHPG (right panel).

In other words, although the non-specific freezing response to both tones is enhanced in the presence of DHPG during the habituation session, enhancement in subsequent learned fear was specific to the CS^+^ only (i.e. higher cue-specific and not generalized fear). This indicates that it is the tone-shock pairing in the presence of DHPG (the cellular equivalent of this process is depicted in the top panel of Figure 3) that boosts the subsequent cue-specific fear learning, not the pre-exposure to tones. In fact, this is consistent with the study mentioned by the referee (Rudy and Matus-Amat, 2009), wherein CS-US pairing was performed in the presence of DHPG but *without* any pre-exposure – and fear memory was still enhanced. Therefore, we agree with the reviewers’ interpretation and have included a statement, along the lines of the referee’s statement, in the revised manuscript to indicate this clearly:

“DHPG may persistently increase excitability of amygdalar neurons such that pairing CS^+^ with the weak US in the presence of DHPG during the conditioning period, is sufficient to enhance fear learning.” Finally, as indicated by the referee, we have used this revised text to also mention earlier findings on enhanced excitability (priming) in the hippocampus (Cohen and Abraham, 1996; Ireland and Abraham, 2002).

*f) The authors also trained mice for fear learning to specific tones in different contexts. What is the role of the different contexts here? Does DHPG affect context specific fear learnings as well as to tones? This would strengthen the authors' conclusion that mGluR5 promotes discriminate learning.*

In this study we focused only on an auditory version of discriminative fear conditioning in rats. Hence, we used distinct tones in different contexts to minimize the potential contributions of context in fear generalization. This strategy was guided by our earlier finding that amygdalar neurons encode the switch from cue-specific to generalized fear, behavioral measures we use in the present study to examine the effects of in vivo activation of mGluR5 in the same brain area (Ghosh and Chattarji, 2015).

We agree with the referee that a similar effect may exert its influence contextual fear learning too. Based on our findings, it is quite likely that DHPG infusion would lead to immediate increase in indiscriminate fear of contexts, but still enhance context-specific fear 24 hours later. We have revised the Discussion to mention this point based on the referee’s comment (second paragraph).

*g) The authors demonstrate that DHPG increases excitability of amygdalar neurons and so it is unclear if mGluR5 activity specifically is required to promote fear learning and anxiety or if simply increasing excitability will suffice. One way to test this would be to infuse a GABAA blocker into the amygdala during tone habituation, and test if this increases subsequent CS^+^ fear learning and LTP.*

In the present study, the DHPG-induced increase in excitability in amygdalar neurons was manifested is two ways – enhancement in both intrinsic and synaptic excitability. The in vivo impact of DHPG was an indiscriminate and immediate increase in freezing to both context and tones, without affecting subsequent formation cue-specific fear. In other words, though the synaptic plasticity triggered by CS-US association took place in the backdrop of heightened excitability through enhanced action potential firing and enhanced spontaneous synaptic activity, it did not interfere with the effective encoding of cue-specific fear memory. An alternative outcome could have been that the DHPG-induced increase in neuronal excitability also affected the inputs receiving CS^-^ information in a manner that subsequently increased freezing to the CS^-^ as well (i.e. generalized fear), not just to the CS^+^ alone (i.e. specific fear). Thus, these findings provide a separation between the effects of DHPG on anxiety versus specific fear, as well as neuronal excitability versus associative synaptic plasticity.

We agree that mGluR5 activity is not the only way to mediate such effects. Modulation of GABA_A_R activity can lead to similar effects. Indeed, enhancement of excitability of amygdalar neurons by reducing GABA_A_R activity also has similar effects (Botta et al., 2015) on anxiety and generalization of fear. Further, enhanced excitability of amygdalar neurons also plays a role in enhancement of cue-specific fear memory (Ehrlich et al., 2009; Wolff et al., 2014). Therefore, the suggestion from the reviewer that modulating GABAA activity may also have similar effects is possible. Based on this point raised by the referee, we have revised the Discussion to mention earlier work on the role of amygdalar GABA_A_R in fear and anxiety (third paragraph).

*h) What is the behavior like during the conditioning session on Day 4, and especially towards the end of that session? Is there any evidence that the pretone freezing and the freezing to the CS^-^ start to go away when the associative conditioning begins?*

We thank the referee for raising this interesting question. We have carefully reviewed the data based on this query. The animals show indiscriminate freezing throughout the conditioning session on Day 4. The pretone freezing to CS^-^ does not go away when the associative conditioning begins. Interestingly, there is higher indiscriminate freezing by the end of the conditioning session. We have now summarized these data in Figure 2—figure supplement 2 and the Results have been modified accordingly (second paragraph).